# Rabies Vaccination of 6-Week-Old Puppies Born to Immunized Mothers: A Randomized Controlled Trial in a High-Mortality Population of Owned, Free-Roaming Dogs

**DOI:** 10.3390/tropicalmed5010045

**Published:** 2020-03-12

**Authors:** Sintayehu Arega, Anne Conan, Claude T. Sabeta, Jan E. Crafford, Jeanette Wentzel, Bjorn Reininghaus, Louise Biggs, Andrew L. Leisewitz, Melvyn Quan, Felix Toka, Darryn L. Knobel

**Affiliations:** 1Center for Conservation Medicine and Ecosystem Health, Ross University School of Veterinary Medicine, Basseterre, Saint Kitts and Nevis; 2Agricultural Research Council-Onderstepoort Veterinary Institute, OIE Rabies Reference Laboratory, Onderstepoort 0110, South Africa; 3Department of Veterinary Tropical Diseases, Faculty of Veterinary Science, University of Pretoria, Onderstepoort 0110, South Africa; 4Hans Hoheisein Wildlife Research Station, Faculty of Veterinary Science, University of Pretoria, Orpen 1364, South Africa; 5Mpumalanga Veterinary Services, Department of Agriculture, Rural Development, Land and Environmental Affairs, Thulamahashe 1365, South Africa; 6Department of Production Animal Studies, University of Pretoria, Onderstepoort 0110, South Africa; 7Department of Companion Animal Clinical Studies, University of Pretoria, Onderstepoort 0110, South Africa; 8Center for Integrative Mammalian Research, Ross University School of Veterinary Medicine, Basseterre, Saint Kitts and Nevis

**Keywords:** immunogenicity, mortality, maternally-acquired immunity, sex, nonspecific effects of vaccines

## Abstract

To achieve global elimination of human rabies from dogs by 2030, evidence-based strategies for effective dog vaccination are needed. Current guidelines recommend inclusion of dogs younger than 3 months in mass rabies vaccination campaigns, although available vaccines are only recommended for use by manufacturers in older dogs, ostensibly due to concerns over interference of maternally-acquired immunity with immune response to the vaccine. Adverse effects of vaccination in this age group of dogs have also not been adequately assessed under field conditions. In a single-site, owner-blinded, randomized, placebo-controlled trial in puppies born to mothers vaccinated within the previous 18 months in a high-mortality population of owned, free-roaming dogs in South Africa, we assessed immunogenicity and effect on survival to all causes of mortality of a single dose of rabies vaccine administered at 6 weeks of age. We found that puppies did not have appreciable levels of maternally-derived antibodies at 6 weeks of age (geometric mean titer 0.065 IU/mL, 95% CI 0.061–0.069; *n* = 346), and that 88% (95% CI 80.7–93.3) of puppies vaccinated at 6 weeks had titers ≥0.5 IU/mL 21 days later (*n* = 117). Although the average effect of vaccination on survival was not statistically significant (hazard ratio [HR] 1.35, 95% CI 0.83–2.18), this effect was modified by sex (*p* = 0.02), with the HR in females 3.09 (95% CI 1.24–7.69) and the HR in males 0.79 (95% CI 0.41–1.53). We speculate that this effect is related to the observed survival advantage that females had over males in the unvaccinated group (HR 0.27; 95% CI 0.11–0.70), with vaccination eroding this advantage through as-yet-unknown mechanisms.

## 1. Introduction

The World Health Organization (WHO), the World Organization for Animal Health (OIE), the Food and Agricultural Organization of the United Nations (FAO), and the Global Alliance for Rabies Control (GARC) have endorsed a plan to eliminate human deaths from dog-mediated rabies by 2030 [1]. A key short-term outcome of the Global Strategic Plan to End Human Deaths from Dog-Mediated Rabies by 2030 is to develop evidence-based tools and strategies for effective dog vaccination [2]. Dogs younger than 3 months often comprise sizeable proportions of the dog populations in low- or middle-income countries and economic models suggest that these puppies should be included in vaccination campaigns to achieve optimal rabies control [3], but consensus on evidence-based approaches for effective vaccination of this segment of the population is lacking.

Rabies vaccines recommended by the WHO and the OIE for the control of rabies in dogs in endemic areas are injectable inactivated cell-culture vaccines [4,5]. Primary vaccination with these vaccines is recommended by manufacturers as a single dose at 12–14 weeks of age, followed by a repeat dose one year later and subsequent revaccination every three years. The basis for recommendation of primary vaccination no earlier than 12 weeks is not clear, but may be an assumption of inhibitory effects of maternal antibodies (which was a concern for the efficacy of earlier live rabies virus vaccines), or the potential for interference of maternal antibodies with interpretation of serological test results for regulatory purposes. In the context of mass dog rabies vaccination campaigns, international and national guidelines allow for adaptation of vaccination schedules from those recommended by vaccine manufacturers, including vaccination of dogs younger than 12 weeks without regard for maternal immune status [4,5,6]. Evaluation of the effect of maternal immune status on the immunogenicity of the vaccine in this age group is important in the context of increased dog vaccination in the Zero by 30 strategic plan, as more puppies will be born to vaccinated dams (mothers).

The adverse effects of the rabies vaccine in puppies younger than 12 weeks under field conditions have also not been adequately addressed. Although Morters et al. [7] state that no adverse events were reported in their field study of rabies vaccination of puppies in resource-poor settings in South Africa and Tanzania, they only considered injection site reactions (granulomas and sterile abscesses) and death on the day of vaccination as adverse events. Of the 68 puppies originally vaccinated in the South Africa site in their study, 61 (90%) were dead or unaccounted for 90 days later. The absence of a control group in their study design precludes an assessment of a causal relationship between vaccination and death as an adverse event. In children in high-mortality settings, increases in all-cause mortality rates have been reported in response to the diphtheria-tetanus-pertussis (DTP) vaccine [8,9], and this effect is reportedly more pronounced in females [10]. This topic remains controversial [11], as much of the evidence is from studies with high risk of bias conducted in a limited range of settings [12,13]. The RTS,S/AS01 malaria vaccine—another nonlive vaccine—is also associated with higher all-cause mortality in female but not male children [14,15]. More generally, both safety and immunogenicity of vaccines are shown to be influenced by sex [16,17,18,19], and greater consideration must be given to sex as a biological variable in vaccine studies. It has also been proposed that sex-differential nonspecific effects of vaccines on survival in early life may be influenced by maternal immune status [20].

The primary objectives of our study were to compare, in owned puppies born to previously-vaccinated dams in a high-mortality population, the antibody response and survival rates between puppies that received an injection of rabies vaccine at 6 weeks of age, and puppies that received a placebo injection of sterile water at the same age. The co-primary endpoints for antibody response (immunogenicity) were geometric mean rabies virus neutralizing antibodies (RVNA) titers and proportion of RVNA titers ≥0.5 IU/mL (seroresponse proportion), measured 21 days after vaccination [21]. The primary endpoint for survival analysis was death due to any cause over 7 weeks of follow up (through 12 weeks of age). As a secondary objective, we assessed the effect of sex on immunogenicity and survival.

## 2. Methods

### 2.1. Study Design

The study design was a single-site, owner-blinded, randomized, placebo-controlled, two-sided comparative study. To remove variation due to maternal immune status, we elected to restrict enrolment to puppies born to dams vaccinated within the 12 months prior to the start of the study. At 6 weeks of age, puppies within litters were randomly assigned in a 1:1 ratio to either a control group or a treatment group. The control group received a subcutaneous injection of sterile water (SW) and the treatment group received a subcutaneous injection of rabies vaccine (RV; Defensor^®^ 3, Zoetis, South Africa). Puppies were followed for up to 7 weeks. At 13 weeks of age, the remaining puppies in both groups received an injection of RV. The study was approved by the Institutional Animal Care and Use Committee of Ross University School of Veterinary Medicine (RUSVM), the Animal Ethics Committee of the University of Pretoria (UP), the Institutional Review Board of RUSVM and the Department of Agriculture, Forestry and Fisheries of South Africa. Written informed consent was obtained from dog owners to participate in the study.

### 2.2. Study Population

The source population for the study was a population of owned, free-roaming dogs in a resource-poor community in South Africa. This location is the site of a health and demographic surveillance system in dogs [22]. Adult female dogs resident within the demographic surveillance area (DSA) were eligible if they were confirmed to be 12 months of age or older by the owner, and if they had been vaccinated against rabies any time in the preceding 12 months, confirmed by a valid vaccination certificate. Unvaccinated dogs or those whose owners could not produce a valid vaccination certificate were vaccinated by the state veterinary technicians or by the study team at the time of recruitment. Owners provided signed informed consent if they were willing to allow the female dog and any future offspring to participate in the study. Female dogs were followed up until parturition.

### 2.3. Treatment Group Allocation and Blinding

Litters were excluded from the study if the date of birth could not be established with reasonable accuracy (within ±3 days). Litters were followed up fortnightly until puppies were 6 weeks (42 days) old. If puppies changed ownership within the DSA during this period, the new owner was approached for enrollment of the puppy into the study. At 6 weeks of age, puppies, stratified by litter, were randomly allocated in a 1:1 ratio to receive a subcutaneous injection of either RV (1 mL of Defensor^®^ 3, Zoetis, South Africa; batch numbers 139280, 139580, 164036, 152142, 170030, 221700, 230361) or placebo (1 mL of SW). Puppies were excluded if the allocation and intervention could not be done within ±3 days of 42 days of age. All puppies remaining in the study received a subcutaneous injection of RV at 13 weeks of age.

Prior to allocation, puppies were weighed and a blood sample collected (<5% of circulating blood volume, estimated as 85 mL/kg). Randomization was done by randomly allocating a treatment group (vaccine or placebo) to each of the series of radio-frequency identity microchips (each with a unique number) to be implanted in the puppies in the litter. One person in the study team, without knowledge of the treatment allocation, would select a microchip and implant it in a puppy. They would read the microchip number to a second person, who would refer to the allocation form of the microchip series for the treatment group code. The first person would select a preprepared syringe labeled with the treatment group code and inject it subcutaneously into the puppy. This process ensured allocation concealment and blinding of the owner to group allocation.

### 2.4. Baseline Characteristics

Owners were interviewed at baseline using a standardized questionnaire to determine type of puppy housing and reported puppy growth. Packed cell volume (PCV) and total protein (TP) were measured on the day of allocation (baseline) according to established protocols. RVNA titers were also measured at baseline (see below).

### 2.5. Follow Up

Puppies were followed up by weekly home visits or phone calls when owners were not at home. If puppies changed ownership within the DSA during this period, the new owner was approached for enrollment of the puppy into the study. Puppies who moved out of the DSA, or whose new owners did not consent to participate, were lost to follow up and censored on the date of exit. Owners were asked to report puppy deaths or disappearances immediately via phone call or text message. The study team visited these households to collect the carcass if available and conduct a verbal autopsy with an adult who was the primary caretaker of the dog [23]. Causes of death were classified as infectious, congenital/birth-related, trauma/accident, and unknown.

### 2.6. Immunogenicity

Blood samples for RVNA titers were collected at three time points: (i) 6 weeks of age (preinjection), (ii) 9 weeks of age (3 weeks after the first injection, and (iii) 16 weeks of age (3 weeks after the second injection. RVNA titers in serum samples were determined by the fluorescent antibody virus neutralization test (FAVN) performed at the Rabies Unit, Agricultural Research Council-Onderstepoort Veterinary Institute, according to the standard procedure described previously [5,24]. Subjects with a postvaccination RVNA titer level of at least 0.5 IU/mL were considered to have demonstrated an adequate response to vaccination, whereas those with a titer below 0.5 IU/mL did not show an adequate response. The titers of the dogs were established by comparison of the reactivity of the test sera to that of the OIE dog reference serum (at 0.5 IU/mL).

### 2.7. Sample Size and Statistical Analysis

The required sample size was estimated for the survival analysis, based on a significance level of 5%, power of 80%, and an effect size (hazard ratio) of 0·33, with a fixed follow-up time of 7 weeks. The probability of death over this period was estimated from the observed mortality rates in the 0−3 month age group from the health and demographic surveillance system [25], assuming exponential survival times [26]. With an expected loss to follow up of 25%, we arrived at a required sample of 354 puppies.

For the immunogenicity data analysis, we report the geometric mean concentration (GMC) of RVNA titers and the proportion of subjects seroresponding for the treatment group, 21 days after first RV injection at 6 weeks. Summary values are reported with 95% confidence intervals (CIs). We also compared these values with those in the control group, 21 days after RV injection at 13 weeks. For the secondary objective of evaluating the effect of sex on immunogenicity of RV, we compared RVNA titers and proportion seroresponding between males and females at both time points (21 days after first RV vaccination in each group). For comparison of RVNA titers, we used the Wilcoxon rank sum test with continuity correction. For comparison of proportion seroresponding, we used Fisher’s exact test. All statistical tests were done with a prespecified alpha of 0.05.

For the survival analysis, effect size was estimated using a mixed-effect Cox proportional hazards model with litter as a random effect, and adjusting for sex and body weight, on the basis of prior knowledge that these variables (selected in advance of the analysis) might have a substantial bearing on survival and therefore improve statistical power [27]. For the secondary objective of evaluating a differential effect of treatment on survival between males and females, we included an interaction term between sex and treatment group. Statistical significance of the interaction term was evaluated using the likelihood ratio test (LRT). To evaluate the robustness of our primary analysis to violation of the assumption of independent censoring, we performed a sensitivity analysis in which we assumed that those puppies reported as lost or stolen were dead. In a second sensitivity analysis, deaths due to accidents were censored.

## 3. Results

Between 3 December 2016 and 8 May 2018, 358 puppies from 87 litters were randomized at 6 weeks of age, 179 to the treatment group and 179 to the control group (Figure 1). All litters were born within 18 months of dam vaccination (median time from vaccination to birth 208 days, ranging from 3 to 555 days). The randomization resulted in balanced groups with regard to baseline demographics, health and owner care (Table 1).

### 3.1. Immunogenicity

RVNA titer results for each subject are shown in Figure 2, and summary results by group and time point are presented in Table 2. Prior to the first injection, at 6 weeks of age, the geometric mean titers (GMT) in both groups combined (*n* = 346) was 0.065 IU/mL (95% CI 0.061–0.069) and the proportion with titers ≥0.5 IU/mL was 9/346 (2.6%, 95% CI 1.2–4.9). Vaccine administered at 6 weeks of age was immunogenic, with 88% (95% CI 80.7–93.3) of puppies showing an adequate response 21 days later (RVNA titers ≥0.5 IU/mL), compared to only 2.8% in the control group. Comparing the response 21 days after primary vaccination at 6 weeks old (treatment group, *n* = 117) with the response 21 days after primary vaccination at 13 weeks old (control group, *n* = 50), there was no statistically significant difference in GMT (1.47 IU/mL vs. 1.18 IU/mL, Wilcoxon rank sum test with continuity correction *p*-value = 0.47), or in proportion seroresponding (88% vs. 84%, Fisher’s exact test *p*-value = 0.47). Providing a booster at 13 weeks in the treatment group increased the GMT from 1.47 IU/mL (95% CI 1.19–1.83) to 2.73 IU/mL (95% CI 1.83–4.06) and the proportion seroresponding from 88.0% (95% CI 80.7–93.3) to 91.8% (95% CI 80.4–97.7).

Although primary vaccination produced a higher GMT and a higher seroresponse proportion in females than males, this effect of sex on immunogenicity was not statistically significant (Table 3).

### 3.2. Survival Analysis

We recorded 80 deaths during the follow-up period from 6 weeks through 12 weeks of age, and 93 puppies (26%) were lost to follow up in this period. Mean follow-up time was 35.0 days (median 47.0, interquartile range [IQR] 22.5–48.0) for the RV group and 33.4 days (median 47.0, IQR 15.0–48.0) for the SW group.

Body weight at 6 weeks of age did not satisfy the proportional hazards assumption, so we categorized the variable using tertiles, producing three strata of body weight (low, medium, high) and then fitted a stratified Cox model, adjusting for sex and with a random effect of litter.

In the primary analysis, the mortality rate was higher in the RV group (2664/1000 dog-years) than in the SW group (1955/1000 dog-years). The hazard ratio (HR) for the average effect of vaccination on survival (adjusted for sex and body weight, with a random effect of litter) was 1.35 (95% CI 0.83–2.18).

The secondary analysis showed substantial modification by sex of the effect of vaccination on survival (LRT *p*-value for interaction term = 0.02), with the HR in females 3.09 (95% CI 1.24–7.69) and the HR in males 0.79 (95% CI 0.41–1.53) (Figure 3, Table 4). It is perhaps pertinent to note that, within the control group, females had a substantial survival advantage over males (HR 0.27; 95% CI 0.11–0.70; derived from Table 4 with control males as reference).

Twenty-two puppies were reported as lost or stolen (eight females and four males in the SW group and eight females and two males in the RV group). Changing these outcomes from censored to deaths in the sensitivity analysis shifted HRs towards the null and decreased standard errors (Appendix A), removing the modification of effect by sex (HR in females 1.71, 95% CI 0.87–3.34; HR in males 0·84, 95% CI 0.46–1.55). Five deaths were classified as accidental (one in the SW group and four in the RV group). With these events censored, the HR in females was 2.83 (95% CI 1.12–7.15) and the HR in males 0.66 (95% CI 0.33–1.34) (Appendix A).

## 4. Discussion

We show that puppies born to dams vaccinated against rabies do not have appreciable levels of maternally-derived antibodies at 6 weeks of age, and that the rabies vaccine is immunogenic when administered to puppies at this age. Data from this study does however suggest that a full dose of vaccine administered at 6 weeks of age decreases survival rates in female but not male puppies.

Chappuis [28] reported that puppies born to nonimmune dams and vaccinated with a killed adjuvanted rabies vaccine at 1 day of age are capable of developing an adequate titer of rabies virus neutralizing antibodies, with this age group developing higher titers than older puppies. In a separate experiment reported in the same paper [28], it was shown that puppies born to dams who received a booster dose of Rabisin^®^ during pregnancy had high titers of maternally-derived antibodies at 14 days of age. Following vaccination with Rabisin^®^ at 14 days of age, titers failed to increase 2 weeks later, and declined significantly thereafter; however, when puppies were challenged with a field canine rabies strain at 120 days of age (at which point the mean antibody titer was low: 0.22 IU/mL), all puppies were fully protected. These data show that under experimental conditions, young puppies are sufficiently immunocompetent and that maternally-derived antibodies do not limit the protective efficacy of inactivated adjuvanted rabies vaccine. In a field study in South Africa and Tanzania, Morters et al. [7] reported that of 27 puppies under 3 months of age vaccinated with Nobivac^®^ Rabies, all seroconverted. Inference regarding the effect of maternal immunity from this study is however limited, as the vaccination status of the dams was uncertain, and prevaccination titers were only determined in 10 of the 27 puppies (GMT 0.09 IU/mL; 95% CI 0.06–0.14).

Our immunogenicity results are similar to those of Wallace et al. [29] who reported results from 290 puppies vaccinated at <12 weeks of age and whose titers were tested by FAVN prior to pet travel: 83.4% had titers ≥0.5 IU/mL, and the GMT was 1.22 IU/mL (95% CI 0.97–1.54). The mean time period between vaccination and blood sample collection in their study was 92 days (95% CI 83–100), longer than our study (21 days), which could account for the slightly lower antibody levels in their study. For the subset of puppies in their study for which titers were tested 15–30 days after vaccination, 94.2% had titers ≥0.5 IU/mL and the GMT was 2.51 IU/mL (95% CI 1.99–3.16). Similar to our study, there was no evidence of a difference in immunogenicity between primary vaccination at <12 weeks vs. at 12–16 weeks. Postvaccination GMTs in our study and that of Wallace et al. [29] were far lower than those obtained by Morters et al. [7] in their sample of 19 dogs <12 weeks of age tested by FAVN 30 days after vaccination (20.68 IU/mL; 95% CI 12.54–34.09). Reasons for this difference are not known but might include differences in vaccines, sites, and routes of administration, differences between testing laboratories, and simultaneous administration of other vaccines or treatments (in the study by Morters et al [7], puppies were simultaneously vaccinated against canine parvovirus and canine distemper, and a proportion received an injection of ivermectin). Regardless, the data from these three studies and that of Chappuis [28] provide convincing evidence that adjuvanted inactivated cell-culture rabies vaccines are immunogenic in dogs younger than 12 weeks irrespective of maternal immune status, and provide support for the WHO and OIE position of adapting manufacturer-recommended vaccination schedules to optimize rabies control.

While the primary analysis of survival data failed to reject the null hypothesis of no difference in survival from 6 through 12 weeks between the vaccinated and control groups, the secondary analysis showed that sex substantially modified the effect of vaccination on survival. Compared to an injection of sterile water, injection of a full dose of rabies vaccine at 6 weeks of age increased the hazard of death in females by more than 3 times over 7 weeks, whereas there was no evidence of an effect in males. Our results are similar to the pattern of increased female (but not male) mortality that has been reported by one group for DTP vaccine in children in high-mortality settings [8,9,30,31,32,33,34,35,36]. A meta-analysis by the same group testing the hypothesis of sex-differential effects of DTP vaccine on all-cause mortality found that DTP vaccination was associated with increased mortality in girls (mortality rate ratio (MRR) 2.54, 95% CI 1.68–3.86) but not boys (MRR 0·96, 95% CI 0.55–1.68) [10]. A WHO-commissioned systematic review provided more equivocal evidence, concluding that receipt of DTP was associated with a possible increase in all-cause mortality on average (relative risk 1.38, 95% CI 0.92–2.08), with the effect seeming stronger in girls than in boys [12]. The safety profile of the RTS,S/AS01 malaria vaccine in infants and children in sub-Saharan Africa also showed that the effect of vaccination on all-cause mortality was substantially modified by sex, with higher all-cause mortality in girls but not in boys [14,15]. These observations have led some scientists to propose that nonlive vaccines—despite providing specific protection against targeted diseases—may have detrimental nonspecific effects (NSEs) on overall morbidity and mortality rates in recipients in high-mortality populations, and that these effects are more pronounced in females [37,38].

The mechanism through which nonlive vaccines may exert sex-specific detrimental NSEs in high-mortality populations is not known, but our data point to an intriguing hypothesis. In our unvaccinated control group, females had a substantial survival advantage over males. This pattern of increased male mortality at young ages is seen in many populations in environments with a high burden of infectious and parasitic diseases [39,40,41]. It has been shown that for many pathogens males are more susceptible to infection and/or severe disease [42,43,44], and that levels of sex hormones (secreted at high levels in infancy as well as in adolescence) contribute to these sex-differential outcomes through their influence on the immune system [42,45,46,47]. In our study population, the marked survival advantage of females at this age is unlikely to be a result of preferential allocation of resources such as food and health care to females, because of the expressed preference of owners for male rather than female dogs [48]. Our results show that this survival advantage of females is eroded following rabies vaccination, leading us to speculate that vaccination increases susceptibility of females (but not males) to these diseases through modulation of the immune response. Alternatively, vaccines could induce immunopathology in females directly, without a mediating effect of other diseases. In humans, females consistently report more frequent and severe local and systemic reactions to vaccines than males [16,17,49]; however, this hypothesis is not consistent with the apparent absence of detrimental NSEs in populations with a low burden of infectious and parasitic diseases [50].

We previously reported results from an observational study in the same population as the current randomized trial [25]. Controlling for the effects of sex and number of dogs in the residence, and stratifying by age, we found that owner-reported rabies vaccination was associated with a lower risk of death from any cause [25]. We recognize that the association seen in that study may have been affected by residual confounding by factors such as owner behavior not controlled for in the analysis. Moreover, Jensen et al. [51] have shown that effect estimates based on recording vaccination status at successive visits, in which information on vaccinations that occurred between visits is updated at the time of the second visit (so-called “retrospective updating”, as we did in the observational study), can lead to considerable bias in vaccine studies, biasing observed mortality rate ratios towards zero (that is, towards a beneficial effect on survival). They noted that the retrospective updating approach can result in estimates of large beneficial NSEs of vaccines, even in the absence of any true effect. These biases could explain the very different conclusions of our two studies.

It is notable that the human rabies vaccine has been associated with beneficial NSEs in children when used as a comparator vaccine in the RTS,S/AS01 malaria vaccine trial [52,53]. Children in the control group who received the rabies vaccine had lower rates of meningitis and cerebral malaria than children who received the RTS,S vaccine. Because this effect was only observed in the older age group of children in which the rabies vaccine was used as the comparator, and not in the younger age group in which a different vaccine (Menjugate) was used as the comparator, it was suggested that this may be due to a protective NSE of the human rabies vaccine [52,53]. Animal and human rabies vaccines are both cell culture-based vaccines inactivated using beta-propiolactone [5,54], but unlike animal rabies vaccines (and unusually for nonlive vaccines), cell culture-based human rabies vaccines are nonadjuvanted. This raises the possibility that the mechanism of action of the detrimental effect of the animal rabies vaccine on female survival noted in our study may be mediated through the effect of the adjuvant component on the immune system. This hypothesis should be tested in further randomized controlled trials.

Although the randomized, owner-blinded, placebo-controlled design used in the current study provides a more rigorous assessment of the true effect of vaccination on survival than our previous observational study [25], the current study is not without its limitations. The study was designed to detect an overall effect of vaccination on survival, and not effect modification by sex. Multiple subgroup analyses inflate the type 1 error rate above 5% [55,56]; however, we conducted only a single, prespecified subgroup analysis for a prerandomization characteristic (sex) based on a formal statistical test for interaction, a method that maintains the rate of false-positive results at 5% [55,56]. Results of our sensitivity analysis show that the study’s conclusion of an effect of vaccination on survival in females is sensitive to measurement bias, if the puppies reported as lost or stolen (and therefore censored in the analysis) had in fact died. Given the implications of our results and the aforementioned limitations, it is important that the study be replicated. Future studies should also attempt to establish sex-specific causes of death between treatment groups; despite efforts in the current study, we were unable to ascertain these through necropsy in most cases, due to the rapid deterioration of carcasses and delayed owner reports of death. The validity of our estimates also relies on the assumption of independent censoring being met; that is, that “within any subgroup of interest, the subjects censored at time t should be representative of all the subjects in that subgroup who remain at risk at time t with respect to their survival experience” [26]. The majority of subjects lost to follow up were given out or sold by owners, but because of the owner-blinded study design, dependence of this decision (to give out or retain puppies) on treatment status would have to be through a mediating effect, for example, a visible change in health status or body condition score due to treatment. This in itself would represent a nonspecific effect of the vaccine. Techniques such as inverse probability-of-censoring weighted estimation can be used to correct for selection bias [57], but these methods require that data are available on potential predictors of censoring (time-varying covariates such as body weight, body condition score, or other visible health parameters), which we did not collect in the current study. Future studies should consider collection of data on these variables and application of these methods.

Recognition of the effect of individual characteristics such as age and sex on vaccination outcomes in people has led to calls for ‘personalized vaccinology’: that “one size and dose might not fit both sexes” [58]. We think that a similar approach of ‘personalized preventive veterinary medicine’ could be of value, even in the context of mass animal vaccination campaigns. If the results of our study are replicated, more personalized approaches could provide avenues for mitigation of adverse effects; for example, testing the efficacy and safety of lower doses of vaccine in females, or different formulations of vaccines for young animals. The effect and mechanism of action of concurrent prophylactic or therapeutic interventions against other infectious or parasitic diseases on vaccine efficacy and safety by sex should also be assessed. Further development of these evidence-based tools for effective dog vaccination will be important for the attainment of the global elimination of human rabies from dogs by 2030.

## Figures and Tables

**Figure 1 tropicalmed-05-00045-f001:**
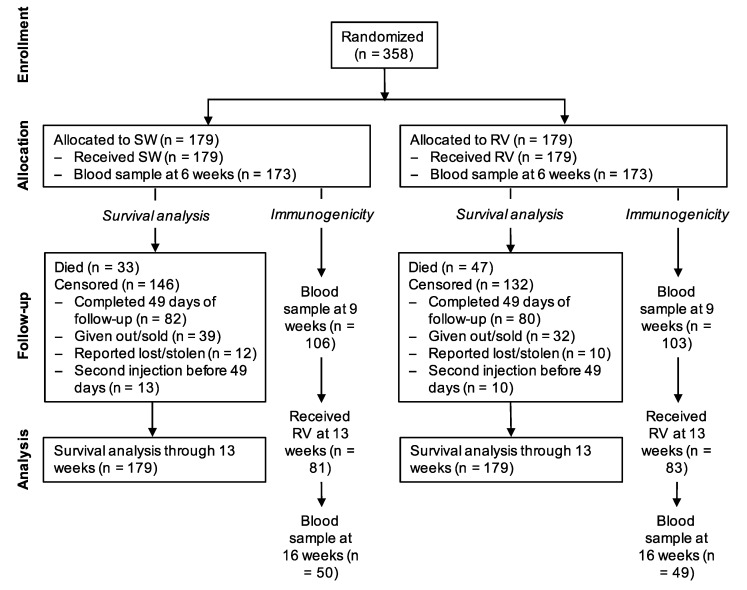
Flowchart of study subjects. SW = sterile water (control); RV = rabies vaccine.

**Figure 2 tropicalmed-05-00045-f002:**
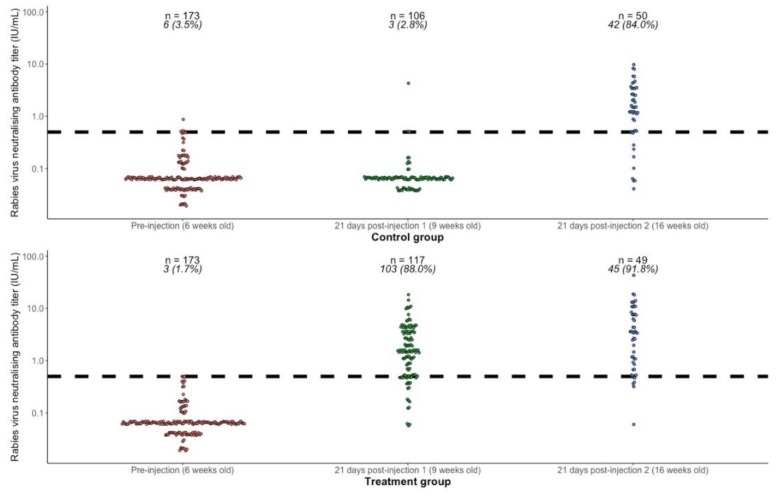
Rabies virus neutralizing antibody titers (RVNA) in the treatment and control groups at 6 weeks of age (prevaccination), 21 days after first injection at 9 weeks of age (rabies vaccine in the treatment group and sterile water in the control group), and 21 days after second injection at 13 weeks of age (rabies vaccine in both groups). The dashed horizontal line shows the threshold of seroprotection (0.5 IU/mL). The numbers above each group show the number of subjects (*n*) and the number and percentage (%) seroresponding (RVNA titers ≥0.5 IU/mL).

**Figure 3 tropicalmed-05-00045-f003:**
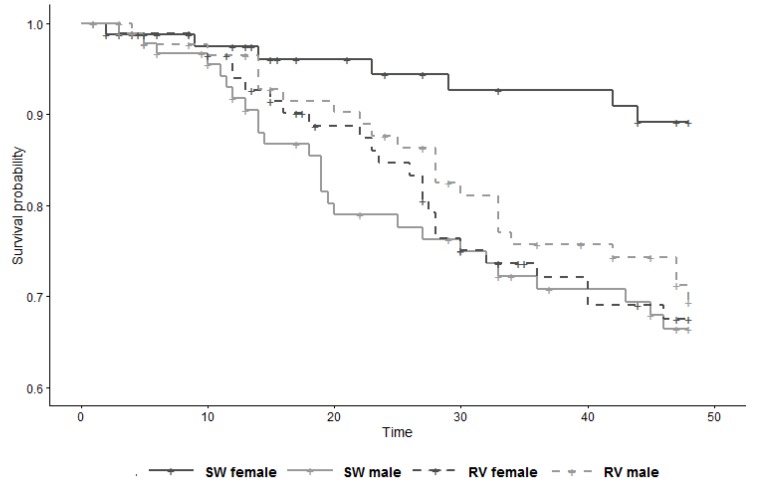
Survival curve of study subjects (*n* = 358) by group and sex, from 6 weeks to 13 weeks of age (time 0 = day of 1st injection). SW = sterile water (control); RV = rabies vaccine.

**Table 1 tropicalmed-05-00045-t001:** Characteristics of subjects at baseline (6 weeks of age).

Characteristic	Sterile Water (SW)	Rabies Vaccine (RV)
Number allocated	179	179
Demographics		
Male, *n* (%)	93 (52.0)	91 (50.8)
Age in days, median (range)	42 (40–45)	42 (39–45)
Dam days between vaccination and birth, median (range) ^a^	208 (3–555)	211 (5–555)
Housing		
Primary caretaker, *n* (%)		
All members of household	97 (54.2)	94 (52.5)
Head of household	57 (31.8)	61 (34.1)
Children	20 (11.2)	16 (8.9)
Other	2 (1.1)	5 (2.8)
No response	3 (1.7)	3 (1.7)
Housing type, *n* (%)		
Mostly inside	116 (64.8)	116 (64.8)
Mostly outside	47 (26.3)	50 (27.9)
Other	13 (7.3)	10 (5.6)
No response	3 (1.7)	3 (1.7)
Health		
Owner-reported growth, *n* (%)		
Normal weight-gain	148 (82.7)	151 (84.4)
Less than expected weight-gain	22 (12.3)	20 (11.2)
Other	6 (3.4)	5 (2.8)
No response	3 (1.7)	3 (1.7)
Weight in grams, mean (sd) ^b^	2037 (699)	2031 (731)
Packed cell volume, mean (sd) ^c^	23.0 (5.4)	22.9 (5.4)
Total protein in g/dL, mean (sd) ^d^	5.2 (0.7)	5.2 (0.7)
RVNA titres in IU/mL, geometric mean (gsd) ^e^	0.07 (1.90)	0.06 (1.79)
Percent with RVNA titres ≥0.5 IU/mL ^e^	3.5	1.7

Notes: gsd = geometric standard deviation; IU = international units; sd = standard deviation; RVNA = rabies virus neutralizing antibodies. ^a^ Assessed in 336 subjects (SW *n* = 168; RV *n* = 168); ^b^ Assessed in 348 subjects (SW *n* = 174; RV *n* = 174); ^c^ Assessed in 240 subjects (SW *n* = 120; RV *n* = 120); ^d^ Assessed in 200 subjects (SW *n* = 100; RV *n* = 100); ^e^ Assessed in 346 subjects (SW *n* = 173; RV *n* = 173).

**Table 2 tropicalmed-05-00045-t002:** Rabies virus neutralizing antibody titers (RVNA) in the treatment and control groups at 6, 9, and 16 weeks of age, showing geometric mean titers (GMT) and percentage seroresponding (RVNA titers ≥0.5 IU/mL).

	Control Group (SW at 6 Weeks + RV at 13 Weeks)			Treatment Group (RV at 6 Weeks + RV at 13 Weeks)		
Age	*n*	GMT in IU/mL (95% CI)	Seroresponse (95% CI)	*n*	GMT in IU/mL (95% CI)	Seroresponse (95% CI)
6 weeks	173	0.066 (0.060–0.073)	3.5% (1.3–7.4)	173	0.064 (0.059–0.070)	1.7% (0.4–4.0)
9 weeks	106	0.064 (0.057–0.071)	2.8% (0.6–8.0)	117	1.47 (1.19–1.83)	88.0% (80.7–93.3)
16 weeks	50	1.18 (0.80–1.74)	84.0% (70.9–92.8)	49	2.73 (1.83–4.06)	91.8% (80.4–97.7)

**Table 3 tropicalmed-05-00045-t003:** Rabies virus neutralizing antibody titers (RVNA) by sex 21 days after primary vaccination with rabies vaccine, showing geometric mean titers (GMT) and percentage seroresponding (RVNA titers ≥0.5 IU/mL).

Primary Vaccination Timepoint	GMT in IU/mL (95% CI)			Seroresponse (95% CI)		
	Females	Males	*p*-Value	Females	Males	*p*-Value
6 weeks (treatment group)	1.53 (1.15–2.05)	1.42 (1.03–1.96)	0.71	89.1% (77.8–95.9)	87.1% (76.1–94.3)	0.78
13 weeks (control group)	1.51 (0.93–2.45)	0.90 (0.48–1.70)	0.20	88.5% (69.8–97.6)	79.2% (57.8–92.9)	0.46

**Table 4 tropicalmed-05-00045-t004:** All-cause mortality from 6 weeks to 13 weeks of age by group allocation and sex for 358 puppies randomly allocated to receive subcutaneous injection of sterile water (*n*= 179) or rabies vaccine (*n* = 179) at 6 weeks of age.

	Sterile Water		Rabies Vaccine		HRs (95% CI) for RV within Strata of Sex
	Mortality Rate ^a^ (Deaths/Dog-Years)	HR (95% CI) ^b^	Mortality rate ^a^ (Deaths/Dog-Years)	HR (95% CI) ^b^	
Females	844 (7/8.3)	1 (reference)	2813 (24/8.5)	3.09 (1.24−7.69)	3.09 (1.24−7.69)
Males	3030 (26/8.6)	3.67 (1.43−9.39)	2525 (23/9.1)	2.89 (1.18−7.11)	0.79 (0.41−1.53)

Notes: HR = hazard ratio. CI = confidence intervals; ^a^ Mortality rate per 1000 dog-years; ^b^ Compared to reference category (control females).

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
