# Peer review of "Rabies Vaccination of 6-Week-Old Puppies Born to Immunized Mothers: A Randomized Controlled Trial in a High-Mortality Population of Owned, Free-Roaming Dogs"

_tropicalmed, 2020, doi:10.3390/tropicalmed5010045_

Round 1

Reviewer 1 Report

The paper by Argea et al. reports a randomized study investigating antibody response and survival rates in puppies following rabies vaccination.  Given the importance of including puppies in mass dog vaccination campaigns for the elimination of human rabies, the paper is of interest. 

The paper is written in good English and presents some interesting results that are worthy of publication.  My primary concerns are with the style of the writing and presentation of the methods and results which I think can be improved to make the paper much easier to follow and a more enjoyable read.  In particular, careful use of well titled sub-headings will aid comprehension.

I will go through my points line by line:

84: put space between the hyphens, or use  commas to separate the clause ‘…in owned…population’

Methods: I felt that renaming of the subsections would help the flow. If possible, I think having a separate subsection for each of the various analyses would help and then have the same subsection headings in the Results.  This would greatly help comprehension.

Also, the authors could try to disentangle the randomization / allocation text from the pure methodological text. At present (as mentioned below) it’s a bit mixed up and this makes following the text difficulty. 

96: Need a short explanation as to why the study was restricted to puppies born to dams vaccinated within 12 months

107: the study didn’t take place within a population of owned dogs.

107: ‘This is the site… ‘ this sentence doesn’t parse well.

110: eligibility criteria: shouldn’t the bitch being pregnant be included in this?

110: add detail explaining how vaccination in past 12 mo was determined

115: this section is not exclusively about randomization and blinding, suggest rename. Eg. Text about treatment of each group is included in line 120, 124

125: interesting that randomization was not stratified by litter.  As a result I suspect there were litters in which one treatment arm was over represented.  Or the effect of between litter variance (which is likely to be high) should have been included in the sample size calculations.  Some text explaining this would help.

133: Study Outcomes – this section includes a lot of study design and suggest this section should be renamed.

135: delete ‘which was SW…..group)’ as this has been explained.  Similarly, ‘(3 weeks….groups)’ should be moved to previous section (eg 120) where describe treatment of each group.

147: ‘Owners were…’ this is a new subject and should be a new paragraph.

148-9: health parameters – what HPs?  Need detail.  And ‘verified by assessment’ – need detail on how this verification was performed.

149: ‘RVNA titres…’ again this is a new subject and should be a new paragraph

158: how was the effect size of 0.33 estimated?

172-81: It would help comprehension if the models were shown

Results: I found the results hard to follow and frequently had to go back and re read bits to ensure I was following the meaning.  As with the methods, this could be improved with judicious use of subsection headings as suggested above.

201: ‘In both groups was 0.065..’  - in T2 its 0.064 and 0.066

203: need to report response in the control group too.

217: Fig 2: x-axis label: Experimental Group – change to Treatment Group (as used in Mthods)

228-231: Are these results shown in a table? If not, then would help if they were.

232-233: ‘In the primary ….dog years)’.  Should mention that this diff was insignificant (as mentioned in Discussion 296).  Also are these primary analysis survival results shown in a survival curve? If not I think they should be.

235: ‘The secondary ….’ this is a new subject and should be a new paragraph.

236:  Do the hazard models involve regression analyses – I guess we would know this if the models were explicitly shown as suggested above).  If so, the output from the hazards model analyses should be shown in full, including all explanatory terms and coefficients etc. Or is T4 the full output?

235-237: The secondary analysis would benefit from being presented in a separate table from Table 4. Or change the layout / formatting as currently its hard to follow.

232-239: I found this paragraph confusing and had to read it several times to fully understand it. Separating the results of the secondary analysis from T4 as suggested above would help, plus perhaps describing all of the HR results in turn might help. Eg. The HR results between male and female for SW should be described; the same for rabies vaccine group; before moving on to the secondary analysis.

240: What sex were the 22 pups that were lost? Needs to be reported as the sex ratio may be skewed and might explain why, when these outcomes are shifted to deaths, the HRs are shifted towards the null.

260-271: ‘These data show ….rabies vaccine.’ To be precise, these results as described in the the Discussion don’t show that maternally derived antibodies do not limit protective efficacy of the rabies vaccine.  They just show that the dogs were immunocompetent. I haven’t read the cited paper for some time, and it may show what the authors are suggesting, but the summary text used in this paper to describe the findings does not illustrate this. 

287: Another important reason not included are lab differences.  i.e differences in the testing labs may explain the differences in the results.

298: ‘…sex substantially modified the effect…’ suggests causality, which has not been shown. Its an association.

329: Females of which species consistently report? Presumably humans, but probably worth specifying.

374-78:  Regression analysis would be useful to see if lost to follow up was determined by treatment group.

Reviewer 2 Report

Overall this is an interesting and useful study regarding the immunogenicity of rabies vaccination in young puppies. The data obtained related to the immunogenicity of rabies vaccination appears to be robust and will make a valuable addition to the literature. Even with the substantial losses of subjects over time the rabies data appears robust enough to overcome this limitation. 

The primary concern is in regards to the interpretation of the survival data and the conclusions that are made based upon that interpretation. While it is appreciated that the study was designed using a population potentially at high risk of exposure to rabies these free roaming puppies were also at high risk of death due to other causes or loss to follow up. The authors cite Morters et al., which appears to be a study undertaken in a similar setting. Unfortunately that study reported that 90% of the enrolled puppies were dead or lost to follow up within 90 days. Yet in the study reported here the power calculation was based upon an expected loss to follow up of only 25%, when in fact it was 50%. This assumption is based upon references 25 and 26 but no details are provided, and does not appear to consider the results reported by Morters et al. When analyzing Figure 1 there are 112 puppies listed as lost to follow up in the box labelled 'Excluded' but these do not appear to be mentioned anywhere else. If they were not randomized then why was follow up necessary? Of greater concern, only 82 puppies completed the full follow up period of 49 days in the SW group, and only 80 in the RV group. These numbers represent substantially less than half of each of the original groups. And yet in Figure 1 it is clearly stated that 179 puppies were considered in each group for the survival analysis through 13 weeks. This does not appear to accurately reflect the number of individuals for which data was available. Indeed, as stated in lines 224-225 173 puppies died or were lost to followup from six to 12 weeks of age. Given that half of these losses were not due to death (or at best their fates were unknown) the remaining data set no longer meets the assumptions of the power analysis. For that reason all of the conclusions related to the survival analyses are suspect. As the authors state on lines 237-240 simply changing how some of these cases are attributed in the analysis completely changes the interpretation (see supplementary table S1). In summary, given that the hazards faced by the studies in this population were many, varied and severe it is not reasonable to attribute the increased risk of death in certain subsets of subjects to rabies vaccination alone. It is clear that the reduced rate of death in the females in the SW group may have been due to random chance, and far larger study cohorts would be required in order to address these concerns. It is appreciated that the authors have mentioned these concerns in lines 357-363 and acknowledged that this study was not designed to detect modification of the effect on survival associated with gender. However, due to these valid limitations the conclusions stated on lines 38-41 and the section of the discussion from lines 292-376 are rendered entirely conjectural. 

Reviewer 3 Report

I have no concerns with the manuscript. There are some minor formatting changes especially with the decimal point between numbers and the survival curve figure was missing. I commend the authors for this important contribution. 

Reviewer 4 Report

This is a very well written article that provides important evidence in the context of an important global and tri-partite agenda to increase the mass vaccination of domestic dogs to eliminate human deaths due to dog-mediated rabies by 2030. The authors conducted an appropriately controlled randomized and blinded study, to test the safety of rabies vaccination for puppies that are younger than recommended by the manufacturer’s label (i.e., younger than three months old) and especially given the recommendations that dogs of all ages should be included in mass-vaccination activities. The study methods are very clearly written or portrayed in conceptual figures and are appropriately justified. The authors have provided the key supporting data and their conclusions are balanced and supported by the data and rigorous analyses employed. The key limitations are recognized and the authors rightly encourage additional studies to test whether the finding is reproducible, which could be possible from the methods provided. I have only a few minor suggestions to improve clarity.

L100-101 – can the authors provide rationale for the duration of follow up – i.e., why was 7 weeks chosen in advance? Certainly, the duration of follow up would be expected to affect the results of the survival analysis in a high-mortality population?

L147-149 – can the authors comment on whether the owners were asked to report whether their dog(s) may receive or did receive any other veterinary vaccines (e.g. parvo, distemper) or treatments? It is important in terms of providing a complete baseline or treatment status of the dogs used in this study.

L150-151 – please specify here also that the criteria to evaluate whether an animal responded to vaccination was whether its titer ever exceeded 0.5 IU/mL. Please clarify the limit of quantitation or cutoff to distinguish negative versus positive (i.e., below the 0.5 IU/mL level) levels of RVNA. Although the methods of the current study do not deviate from the global standard by qualifying vaccination response based on threshold of 0.5 IU/mL, some studies also qualify responses based on a four-fold rise in titers over the baseline value. Did all responders also show evidence of a four-fold rise in titer at day 21 pv? Perhaps the authors could clarify briefly.

L199 – there is no globally agreed upon protective level of antibody. The authors may consider a word change to simply state what is meant by the 0.5 IU/mL cutoff – that is it is evidence of adequate antibody response to rabies vaccination.

L228-231 – the authors should either specify that these (combined sex) results are not shown, or include a separate table – i.e., hazard ratios determined without stratification based on sex. The authors should also report the covariance parameter estimate for the random effect of litter (i.e., evidence of litter-specific effects on mortality rate) for each model tested.

L303 – Given the sensitivity of the conclusions to how mortality or censoring observations are coded, I think the authors could briefly outline the rationale and assumptions behind evaluations of all-cause mortality in human or animal vaccine studies – that is, considering only broader all-cause mortality may under-emphasize biologically important phenomena (immune system modulation) from an analytical standpoint by giving equal weight to mortality events such as accidental death, which may have nothing to do with vaccine or vaccination event. Presumably, this is also why the authors conduct subsequent ‘sensitivity analyses’ on the survival response data. While it can be important to follow the industry standard for a vaccine study of this type, the authors rightly acknowledge that there can be logistical challenges to applying this type of design to a free-roaming high-mortality owned dog population. I commend the authors for considering alternative explanations of their data.

L322-323 – perhaps, but these biological factors or mechanisms are not specifically what was observed or evaluated in this study. The authors concede elsewhere in the article that some events may be inaccurately assumed, recorded or coded for mortality versus censoring (e.g. lost, stolen, accidental death), which are inherent difficulties with patient follow up in a high-mortality setting coupled with an underlying process of shifting animal ownership. Different coding can impact the survival analysis, as shown in this study, where all-cause mortality includes intrinsic and extrinsic factors. There did not seem to be evidence of an increased mortality rate associated due to intrinsic or biological factors in this study, although a few clarifying remarks on this point in the discussion may be beneficial. The intrinsic factors would seem to be the most concerning effects from the standpoint of scaling up mass-vaccination activities which practically may include this very young cohort of animals. I also can appreciate the author’s remarks about the difficulty in determining a cause of death in most animals due to delays in owners reporting the event or condition of the patient tissues for pathogen testing. The discussion of the survival analysis results focuses more on the sensitivity of the results to variable coding of extrinsic factors which, while important to be transparent about, distracts the reader from the priority biological question at hand (are veterinary rabies vaccines generally safe for puppies less than 3 months old). The authors might consider light re-organization of the discussion around some of these points.
